# Short-Term Air Pollution Exposure and Risk of Acute Exacerbation of Chronic Obstructive Pulmonary Disease in Korea: A National Time-Stratified Case-Crossover Study

**DOI:** 10.3390/ijerph19052823

**Published:** 2022-02-28

**Authors:** Yun Jung Jung, Eun Jin Kim, Jung Yeon Heo, Young Hwa Choi, Dae Jung Kim, Kyoung Hwa Ha

**Affiliations:** 1Department of Pulmonary and Critical Care Medicine, Ajou University School of Medicine, Suwon 16499, Korea; tomato81@aumc.ac.kr; 2Department of Infectious Diseases, Ajou University School of Medicine, Suwon 16499, Korea; 111115@aumc.ac.kr (E.J.K.); jyheomd@aumc.ac.kr (J.Y.H.); yhwa1805@ajou.ac.kr (Y.H.C.); 3Department of Endocrinology and Metabolism, Ajou University School of Medicine, Suwon 16499, Korea; djkim@ajou.ac.kr

**Keywords:** air pollution, acute exacerbation, chronic obstructive pulmonary disease, insurance claims

## Abstract

We investigated the association between short-term exposure to air pollution and the risk of acute exacerbation of chronic obstructive pulmonary disease (AE-COPD) in seven metropolitan cities in Korea. We used national health insurance claims data to identify AE-COPD cases in 2015. We estimated short-term exposure to particulate matter (PM) with a diameter of ≤2.5 μm (PM_2.5_), PM with diameters of ≤10 μm (PM_10_), sulfur dioxide (SO_2_), nitrogen dioxide (NO_2_), carbon monoxide (CO), and ozone (O_3_) obtained from the Ministry of Environment. We conducted a time-stratified, case-crossover study to evaluate the effect of short-term exposure to air pollution on hospital visits for AE-COPD, using a conditional logistic regression model. The risk of hospital visits for AE-COPD was significantly associated with interquartile range increases in PM_10_ in a cumulative lag model (lag 0–2, 0.35%, 95% confidence interval (CI) 0.06–0.65%; lag 0–3, 0.39%, 95% CI 0.01–0.77%). The associations were higher among patients who were men, aged 40–64 years, with low household income, and with a history of asthma. However, other air pollutants were not significantly associated with the risk of hospital visits for AE-COPD. Short-term exposure to air pollution, especially PM_10_, increases the risk of hospital visits for AE-COPD.

## 1. Introduction

Chronic obstructive pulmonary disease (COPD) is a progressive inflammatory disease characterized by airway limitation that is not completely reversible [1]. According to the World Health Organization (WHO), COPD is the third leading cause of death worldwide [2]. Acute exacerbation of COPD (AE-COPD) is defined as acute worsening of respiratory symptoms, such as dyspnea, cough, and sputum production, beyond normal day-to-day variations, resulting in changes in regular medication [3]. AE-COPD has a significant impact on a patient’s quality of life and can accelerate the decline in lung function [4]. Thus, AE-COPD is a major cause of morbidity, mortality, and burden on public health services [5,6].

The causes of AE-COPD are multifactorial. Common causes include respiratory infections (bacterial and viral infections), exposure to smoking, and poor drug adherence [7,8,9]. Additionally, air pollution may also cause exacerbation. Several studies have reported an association between exposure to air pollution and increased emergency room (ER) visits and hospitalizations due to AE-COPD [10,11,12]. However, the adverse effects of air pollution on AE-COPD remain controversial. Moreover, several studies have reported on the effects of air pollution on AE-COPD in Korea; however, all of them were conducted in a single hospital, a single city, or a local area [13,14,15].

Thus, we investigated the association between short-term exposure to air pollution and the risk of AE-COPD in seven Korean metropolitan cities in 2015, using a nationwide health claims database.

## 2. Materials and Methods

### 2.1. Hospital Visits for AE-COPD

Individual-level data on hospital visits for AE-COPD were collected from seven metropolitan cities in Korea (Seoul, Incheon, Daejeon, Daegu, Ulsan, Gwangju, and Busan) between 1 January 2015 and 31 December 2015 using the National Health Insurance Service–National Sample Cohort (NHIS–NSC). In Korea, the National Health Insurance Service (NHIS) has been providing a single-payer healthcare insurance service to the entire population since 1989. The NHIS established the NHIS–NSC, which consists of 2.2% of eligible Koreans using a systematic sampling method [16].

We defined AE-COPD based on hospital visits (outpatient, ER, or inpatient visits) for COPD, accompanied by a prescription of systemic steroids and/or antibiotics as rescue medications [3,17,18,19]. COPD was defined as a diagnosis of COPD (International Classification of Disease–10th Revision [ICD-10] codes, J43–J44, except J43.0) and at least one of the COPD medications used at least twice within a year. COPD medications include long-acting muscarinic antagonists (LAMAs), long-acting beta-2 agonists (LABAs), LAMA in combination with LABA (LAMA/LABA), inhaled corticosteroids (ICS), ICS in combination with LABA (ICS/LABA), short-acting muscarinic antagonists, short-acting beta-2 agonists, systemic methylxanthines, systemic corticosteroids, and systemic beta-agonists [20,21,22,23]. The rescue medications were defined as follows: (1) systemic steroid (Anatomical Therapeutic Chemical (ATC) codes H02AB and H02BX except for H02AB08 and H02AB13), and (2) antibiotics (ATC code J01). In patients who had been taking systemic steroids, we included cases in which the steroid dose was increased. Hospital visits within 30 days were considered a single event, while hospital visits after 30 days were considered separate events. Patients were excluded if they had the following immunodeficiency disorders or autoimmune diseases: cancer (ICD-10 codes C00–C97) in the preceding 3 years; rheumatic disease (ICD-10 codes M05, M06, M32, M34, M35, and M45); inflammatory bowel disease (ICD-10 codes K50 and K51); transplantation (ICD-10 codes T86 and Z94) in the previous year. The study protocol was reviewed and approved by the Institutional Review Board of Ajou University Hospital (No. AJIRB-MED-EXP-21-222).

### 2.2. Air Pollution and Meteorological Data

Data on daily air pollution collected from local monitoring sites by the Korean Ministry of Environment were analyzed. Particulate matter (PM) with a median aerodynamic diameter of ≤2.5 μm (PM_2.5_), PM with a median aerodynamic diameter of ≤10 μm (PM_10_), sulfur dioxide (SO_2_), nitrogen dioxide (NO_2_), carbon monoxide (CO), and ozone (O_3_) were the forms of air pollution examined. First, we selected 126 monitoring sites (Seoul, 40; Incheon, 18; Daejeon, 10; Daegu, 13; Ulsan, 15; Gwangju, 9; and Busan, 21), which collected data representative of the levels of air pollution in the general population in the seven metropolitan cities. Next, we computed the daily mean of the hourly density of PM_2.5_, PM_10_, SO_2_, NO_2_, CO, and O_3_ for each site, and daily area values were calculated by averaging the daily means of the selected monitoring sites for each pollutant. Data on daily meteorological covariates measured at a fixed-site station in 2015 were collected from the National Meteorological Office. Meteorological variables included mean temperature, dew point temperature, and atmospheric pressure (hPa), and these variables were used as confounders by applying the natural cubic spline function of the 3-day average temperature of the previous three days with three degrees of freedom.

### 2.3. Statistical Analysis

A time-stratified, case-crossover approach was used to analyze the association between short-term changes in air pollutants and the risk of hospital visits for AE-COPD. The approach was used to compare exposure to air pollution before the event (case period) with exposure at other times (control period) in order to control for confounding factors such as individual characteristics [24]. The case period for each hospital visit for AE-COPD was defined as the date of a hospital visit for AE-COPD. To eliminate the potential confounding effects of long-term trends and seasonality, the control period was identified by matching the day of the week within the same year and month, allowing for the inclusion of three or four control days per case day. A two-stage hierarchical model was used to assess the association between short-term exposure to air pollutants and the risk of hospital visits for AE-COPD. In the first stage, a conditional logistic regression analysis, which was adjusted for the nonlinear confounding effects of meteorological factors and national holidays, was performed to estimate the effects of exposure to air pollutants on hospital visits for AE-COPD in each city. After building a core model for each city, different lag patterns were considered using a single- or cumulative-lag model to evaluate immediate and delayed effects. The single lag model comprised lag 0 (current day concentration) to lag 3 (three days before the event day). The cumulative lag model comprised lag 0–1, lag 0–2, and lag 0–3 by calculating average pollutant concentrations on cumulative days up to the day of the visit using polynomial distributed lag models [25]. In the second stage, city-specific results of the first stage were combined using a random-effect meta-regression model to estimate the overall effect of exposure to air pollutants across all cities. We performed several subgroup analyses stratified by sex, age (40–64 years, and ≥65 years), household income (low, middle, and high), season (spring, summer, fall, and winter), history of asthma, and the number of AE-COPD occurrences in the previous year using the cumulative lag 3 model. Since *p*-values were not adjusted for multiple tests, they should be used only for exploratory interpretation. All statistical analyses were performed using SAS Enterprise Guide 7.1 (SAS Institute, Cary, NC, USA).

## 3. Results

Among patients with AE-COPD in 2015, 74.6% were men, 71.3% were aged ≥65 years, 89.5% had a prior history of asthma, and 67.7% had at least one AE-COPD event in the previous year (Table 1).

Table 2 summarizes the average air pollutant concentrations by region. The interquartile ranges (IQR) were as follows: PM_2.5_, 17.46 μg/m^3^; PM_10_, 25.50 μg/m^3^; SO_2_, 3.04 ppb; NO_2_, 13.57 ppb; CO, 206.82 ppb; O_3_, 15.37 ppb.

Figure 1 shows the estimated effect of short-term exposure to air pollution on hospital visits for AE-COPD per IQR increase in different single and cumulative lag models. PM_10_ levels were associated with an increased risk of hospital visits for AE-COPD in lag 0–2 (0.35%; 95% confidence interval (CI) 0.06–0.65%) and lag 0–3 (0.39%; 95% CI 0.01–0.77%). When we further adjusted for the wind speed and rainfall in lag 0–2 and lag 0–3, the results were similar to the main analysis (data not shown).

However, PM_2.5_, SO_2_, CO, NO_2_, and O_3_ were not significantly associated with the risk of hospital visits for AE-COPD in either single or cumulative lag models. We performed further analysis using different estimation methods of air pollutants (maximum 1 h daily concentration of SO_2_, NO_2_, and CO and maximum 8 h daily concentrations of O_3_) in lag 0–3. We have confirmed that the results were similar to the main analysis (average 24 h concentrations of SO_2_, NO_2_, CO, and O_3_; data not shown).

Figure 2 presents the association between PM_10_ and the risk of hospital visits for AE-COPD in lag 0–3 stratified by sex, age, household income, seasons, and history of asthma. In random-effects estimates, the significant association with increased risk of hospital visits for AE-COPD was only in men (0.98%; 95% CI 0.20–1.77%), aged 40–64 years (1.24%; 95% CI 0.07–2.43%), low household income (1.07%; 95% CI 0.13–2.02%), and a history of asthma (0.82%; 0.19–1.45%). With regard to season, all seasons except fall had significant associations with an increased risk of hospital visits for AE-COPD.

Based on the number of AE-COPD occurrences in the previous year, patients with more than one AE-COPD event had significant associations with increased risk of hospital visits for AE-COPD (1, 0.29%, 95% CI 0.11–0.47%; ≥2, 0.36%, 0.28–0.44%).

## 4. Discussion

In Korea, short-term exposure to air pollution increases the risk of hospital visits for AE-COPD. An increase in PM_10_ levels was associated with the risk of AE-COPD during cumulative lag 0–2 and lag 0–3 models. Moreover, a stronger association was observed among patients who were men, aged 40–64 years, with low household income, and a history of asthma. However, PM_2.5_, SO_2_, NO_2_, CO, and O_3_ levels were not associated with the risk of hospital visits for AE-COPD.

The adverse effects of PM on AE-COPD have been reported in previous studies [13,14,26,27,28]. In Korea, PM_10_ levels were associated with increased hospitalization for AE-COPD in lag 3 using electronic records from a single hospital [13]. AE-COPD increased in lags 2, 3, and 4 as the concentration of PM_2.5_ increased in a single city in Korea [14]. Here, an increase in PM_2.5_ levels was associated with the risk of AE-COPD, although this association was not significant. In a meta-analysis study of five cohorts in Europe, PM_coarse_ (PM_10_ minus PM_2.5_) increased the prevalence of respiratory symptoms but not PM_2.5_ [26]. PM_coarse_ might play a greater role in triggering AE-COPD than PM_2.5_ because PM_coarse_ (similarly, PM_10_) is mostly deposited in the primary bronchi [27,28]. However, in patient with asthma, an increase in PM_2.5_ levels led to a significantly increased risk of hospitalization for AE-COPD (4.4%; 95% CI 2.2–6.5%; data not shown). Moreover, PM_2.5_ levels were associated with had a higher risk of AE-COPD than PM_10_ levels (0.8% vs. 4.4%; data not shown). Thus, further studies are required to determine the effects of PM on AE-COPD, depending on whether it is accompanied by asthma.

In China, adverse effects from short-term exposure to gaseous pollutants composed of SO_2_, NO_2_, CO, O_3_, as well as PM, have been reported using the public health information center database [28]. Additionally, in a systematic review and meta-analysis, a significant association was observed between short-term exposure to gaseous and particulate pollutants and AE-COPD [29]. However, the association between short-term exposure to SO_2_ and NO_2_ and the risk of AE-COPD was more evident in developing countries than in developed countries. Previous studies have also reported an unstable association of CO and O_3_ with AE-COPD [30,31,32].

The exact mechanisms underlying the acute effects of air pollution on AE-COPD are not well established. Common causes of AE-COPD include respiratory infection, smoking, and poor drug adherence [7,8,9]. Moreover, the mechanism by which air pollution worsens the airway disease is that the inhalation of particles and gases could induce direct damage to airway epithelial cells, reduce mucociliary clearance, and impair macrophage function through abnormal activation of inflammatory cells and intracellular oxidative stress [33,34]. These mechanisms may also worsen the host’s immunity and make them more susceptible to infections, thereby increasing the incidence of AE-COPD.

Identification of potentially susceptible populations is important to prevent the occurrence of AE-COPD. In a Chinese study, women and those aged ≥65 years were more susceptible to air pollution, including PM_10_ and PM_2.5_ [28]. However, we found a stronger association between PM_10_ and men, and those aged 40–64 years, which was different from the Chinese study. Moreover, patients with a low income were also susceptible to PM_10_. Although such patients have COPD, they may have greater exposure to air pollutants while spending more time outside. However, our study had limitations in evaluating the exact exposure of air pollutants to each patient. It is known that COPD patients with asthma are more susceptible to AE-COPD than those without asthma [35]. The same results were found in this study, that is, the risk of AE-COPD for air pollution exposure was increased in patients with COPD and asthma, compared with those without asthma. Moreover, the number of AE-COPD occurrences in the previous year indicated a gradual increase in AE-COPD risk for air pollution exposure in terms of the increase in the number of patients who had AE-COPD events once or more than twice in the previous year (0.29% and 0.36%, respectively).

This study had several limitations. First, the severity of COPD is determined by spirometry and respiratory symptoms, and AE-COPD is more likely to occur in severe COPD [36]. However, we did not consider the spirometry and respiratory symptoms of patients, which limited our ability to evaluate the effects of the severity of COPD on the incidence of acute exacerbation. Second, although the case-crossover design accounted for confounding factors, unmeasured confounding factors such as exposure to smoking and poor drug adherence are still possible. Third, we investigated the average daily air pollutants collected outdoor but not indoors. This limitation is further aggravated by the fact that the measurement of air pollutants by outdoor monitoring sites may not accurately reflect overall personal exposure.

## 5. Conclusions

In conclusion, increased levels of air pollution, especially PM_10_, increased the risk of hospital visits for AE-COPD. Patients with asthma or AE-COPD events in the previous year were particularly vulnerable. Thus, more efforts are needed to reduce air pollution to meet international air quality guidelines, and susceptibility in patients in terms of the prevention and management of AE-COPD should be particularly considered.

## Figures and Tables

**Figure 1 ijerph-19-02823-f001:**
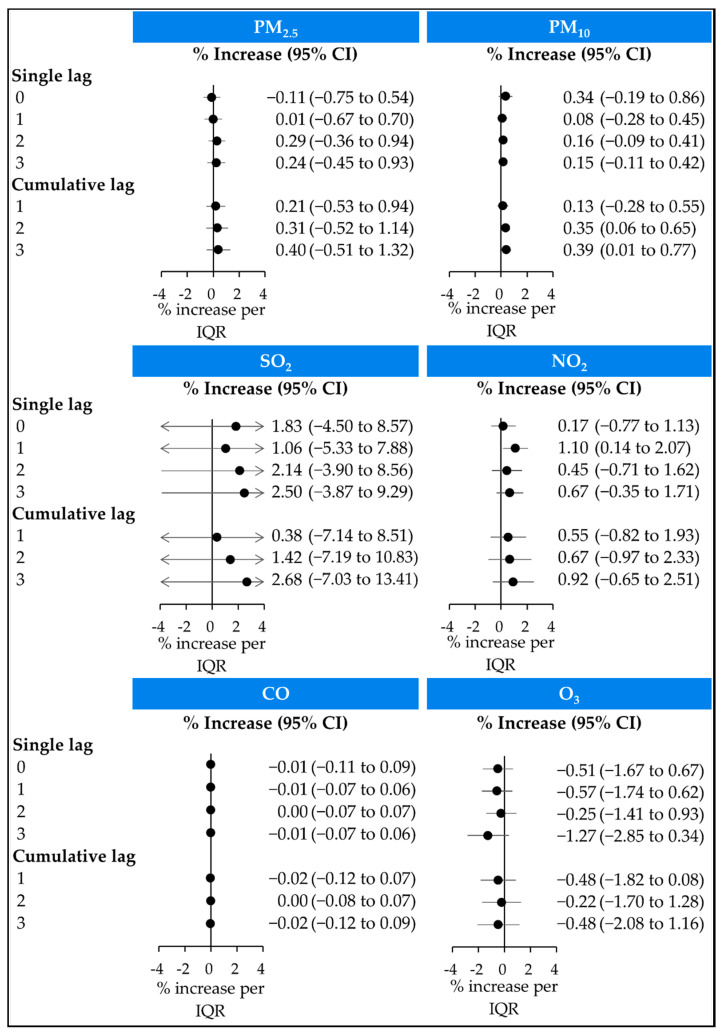
Random-effects estimate of the risk of acute exacerbation of chronic obstructive pulmonary disease associated with an interquartile range (IQR) increase in pollutant concentrations in seven metropolitan cities in South Korea, 2015. The day of outcomes, and 1, 2, and 3 days prior to the day of the hospital visit, were described as lag 0, lag 1, lag 2, and lag 3, respectively. Cumulative lag refers to cumulative lagged exposure, which indicates the mean particulate matter concentrations on a specific day and all subsequent days. After adjusting for national holidays and natural cubic spline variables (mean temperature, dew point temperature, and air pressure), the IQRs were as follows: PM_2.5_,17.46 μg/m^3^; PM_10_, 25.50 μg/m^3^; SO_2_, 3.04 ppb; NO_2_, 13.57 ppb; CO, 206.82 ppb; O_3_, 15.37 ppb. PM_2.5_, fine particulate matter with a median aerodynamic diameter of ≤2.5 μm; PM_10_, fine particulate matter with a median aerodynamic diameter of ≤10 μm; SO_2_, sulfur dioxide; NO_2_, nitrogen dioxide; CO, carbon monoxide; O_3_, ozone; CI, confidence interval. The arrow at the end of the line means that the CI extends over the range of *x*-axis.

**Figure 2 ijerph-19-02823-f002:**
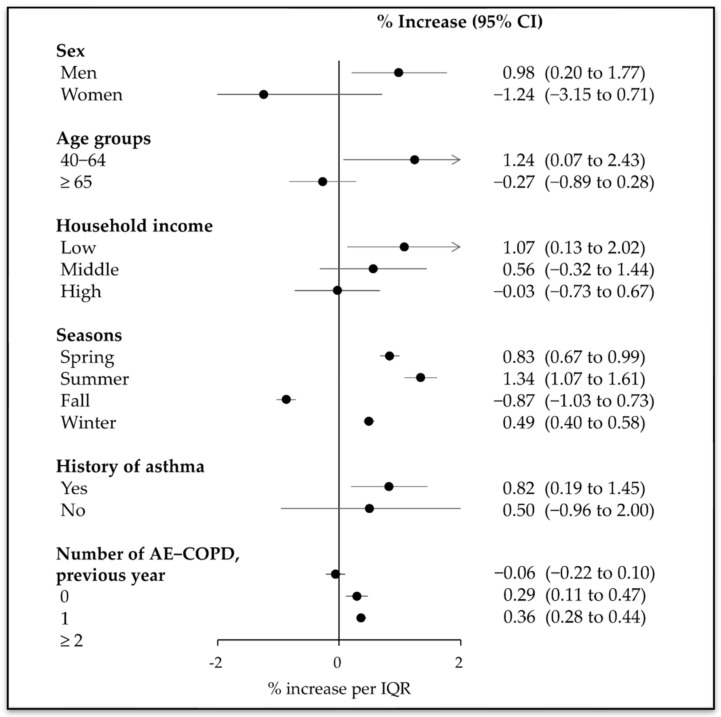
Percent increase in acute exacerbation of chronic obstructive pulmonary disease associated with an interquartile range (IQR) increase in PM_10_ after stratification for various factors (cumulative lag 3). After adjusting for national holidays and natural cubic spline variables (mean temperature, dew point temperature, and air pressure), the IQRs were PM_10_, 25.50 μg/m^3^. PM_10_, fine particulate matter with a median aerodynamic diameter of ≤10 μm; CI, confidence interval; AE-COPD, acute exacerbation of chronic obstructive pulmonary disease. The arrow at the end of the line means that the CI extends over the range of *x*-axis.

**Table 1 ijerph-19-02823-t001:** General characteristics of hospital visits for acute exacerbation of chronic obstructive pulmonary disease in seven metropolitan cities in South Korea, 2015.

Variables	N (%)
Sex	
Men	736 (74.6)
Women	251 (25.4)
Age (years)	
40–64	283 (28.7)
≥65	704 (71.3)
History of asthma	
Yes	883 (89.5)
No	104 (10.5)
Number of AE-COPD, previous year	
0	319 (32.3)
1	173 (17.5)
≥2	495 (50.2)
Household income	
Low	177 (17.9)
Middle	289 (29.3)
High	404 (40.9)
Missing	117 (11.9)
Seasons	
Spring	266 (26.9)
Summer	201 (20.4)
Fall	233 (23.6)
Winter	287 (29.1)
Regions	
Seoul	380 (38.5)
Incheon	163 (16.5)
Daejeon	135 (13.7)
Daegu	107 (10.8)
Ulsan	101 (10.2)
Gwangju	72 (7.3)
Busan	29 (2.9)

AE-COPD, Acute exacerbation of chronic obstructive pulmonary disease.

**Table 2 ijerph-19-02823-t002:** Distribution of air pollutant levels in seven metropolitan cities in South Korea, 2015.

City	Pollutant	10th %Tile	25th %Tile	Median	75th %Tile	90th %Tile
Seoul	PM_2.5_ (μg/m^3^)	10.17	15.83	21.19	28.33	38.87
	PM_10_ (μg/m^3^)	21.41	31.62	41.81	54.84	74.92
	SO_2_ (ppb)	4.34	4.77	5.43	6.30	7.28
	NO_2_ (ppb)	24.80	29.46	36.95	46.84	54.20
	CO (ppb)	393.33	440.71	526.56	682.22	876.02
	O_3_ (ppb)	5.96	11.62	17.86	25.65	32.04
Incheon	PM_2.5_ (μg/m^3^)	12.47	18.77	25.52	36.97	50.69
	PM_10_ (μg/m^3^)	24.94	34.04	44.36	60.03	82.00
	SO_2_ (ppb)	4.20	4.89	5.91	7.02	8.42
	NO_2_ (ppb)	14.36	17.71	23.28	31.36	38.26
	CO (ppb)	276.98	323.65	380.65	452.32	523.15
	O_3_ (ppb)	14.19	20.44	26.79	33.53	39.32
Daejeon	PM_2.5_ (μg/m^3^)	10.00	15.17	24.04	37.38	48.63
	PM_10_ (μg/m^3^)	19.88	28.19	41.90	55.35	74.46
	SO_2_ (ppb)	1.80	2.13	3.06	4.43	5.65
	NO_2_ (ppb)	11.06	13.53	17.30	23.54	30.12
	CO (ppb)	260.70	316.41	391.02	499.48	688.85
	O_3_ (ppb)	7.70	16.03	22.79	33.09	43.90
Daegu	PM_2.5_ (μg/m^3^)	11.58	16.98	23.68	33.98	42.60
	PM_10_ (μg/m^3^)	23.33	31.78	42.63	56.24	75.15
	SO_2_ (ppb)	2.02	2.36	3.07	4.13	5.60
	NO_2_ (ppb)	11.69	15.23	20.03	26.73	34.74
	CO (ppb)	383.48	447.80	526.36	687.84	883.06
	O_3_ (ppb)	13.21	18.82	25.60	35.00	41.10
Ulsan	PM_2.5_ (μg/m^3^)	10.34	14.30	22.52	32.98	43.76
	PM_10_ (μg/m^3^)	23.73	29.73	40.87	54.95	73.45
	SO_2_ (ppb)	4.40	5.20	6.60	9.50	12.81
	NO_2_ (ppb)	13.82	17.22	22.30	28.70	34.33
	CO (ppb)	356.77	416.50	489.65	594.01	724.07
	O_3_ (ppb)	10.91	17.29	25.62	34.08	42.52
Gwangju	PM_2.5_ (μg/m^3^)	11.33	16.23	24.13	33.57	44.81
	PM_10_ (μg/m^3^)	20.43	28.32	40.53	54.00	74.04
	SO_2_ (ppb)	2.36	2.72	3.22	3.88	4.45
	NO_2_ (ppb)	13.29	16.15	19.85	27.58	32.88
	CO (ppb)	303.75	347.29	424.37	570.37	755.55
	O_3_ (ppb)	6.55	13.47	21.93	30.76	40.67
Busan	PM_2.5_ (μg/m^3^)	12.37	16.87	23.88	32.73	41.53
	PM_10_ (μg/m^3^)	24.76	30.40	41.20	54.88	72.32
	SO_2_ (ppb)	4.00	5.10	6.20	7.55	9.17
	NO_2_ (ppb)	12.65	15.63	19.92	25.76	30.50
	CO (ppb)	355.21	410.49	491.00	594.09	687.19
	O_3_ (ppb)	13.37	18.50	24.60	32.48	40.55

PM_2.5_, fine particulate matter with a median aerodynamic diameter of ≤2.5 μm; PM_10_, fine particulate matter with a median aerodynamic diameter of ≤10 μm; SO_2_, sulfur dioxide; NO_2_, nitrogen dioxide; CO, carbon monoxide; O_3_, ozone.

## Data Availability

The data presented in this study are available on request from the corresponding author.

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
