# Peer review of "Short-Term Air Pollution Exposure and Risk of Acute Exacerbation of Chronic Obstructive Pulmonary Disease in Korea: A National Time-Stratified Case-Crossover Study"

_ijerph, 2022, doi:10.3390/ijerph19052823_

Round 1
Reviewer 1 Report
The manuscript presents the association between the risk of acute exacerbation of chronic obstructive pulmonary disease (AE-COPD) with short-term exposure to air pollution in seven metropolitan cities in Korea. The pollutants studied are particulate matter (PM10 and PM2.5), sulfur dioxide, nitrogen dioxide, carbon monoxide and ozone. It results that high level of PM10 increased the risk of hospital visit for AE-COPD.
The work is interesting and well organized. However, some minor revisions are required:
- The meteorological data considered are mean temperature, dew point temperature, and atmospheric pressure. Why not wind and rainfalls? These two variables influence the PM concentrations;
- Table 2: please check subscripts “th”;
- Line 183 pdf version of manuscript “The adverse effects of PM on AE-COPD have been reported in previous studies”: please insert references
Reviewer 2 Report
The authors used 24-hour daily average concentrations of all the air pollutants. If hourly data are available, I wonder if the author can try using different metrics, say 8-hour daily max for O3 and 1-hour daily max for other criteria gases (CO, SO2, NO2). The reason for this suggestion is that these pollutants usually have clear diurnal patterns and thus the 24-hour daily average may not reflect the relevant exposure resulting in health impact. This may be one reason for the null associations.
